# Digital tools for delivery of dementia education for caregivers of persons with dementia: A systematic review and meta-analysis of impact on caregiver distress and depressive symptoms

Andrea Scerbe[1], Megan E. O'Connell[1]*, Arlene Astell[2], Debra Morgan[3], Julie Kosteniuk[3], Ivan Panyavin[1], Andrea DesRoches[1], Claire Webster[4,5]

1 Department of Psychology, University of Saskatchewan, Saskatoon, Saskatchewan, Canada, 2 Department of Psychiatry, University of Toronto, Toronto, Ontario, Canada, 3 College of Medicine, University of Saskatchewan, Saskatoon, Saskatchewan, Canada, 4 Certified Alzheimer Care Consultant, Caregiver Crosswalk Inc., Montréal, Quebec, Canada, 5 McGill University Dementia Education Program, Montréal, Quebec, Canada

* megan.oconnell@usask.ca

**Data Availability Statement:** We have included supporting information as is usual for reviews

## Abstract

Continuing education for dementia has been shown to be beneficial by improving informal caregiver knowledge, dementia care, management, and caregiver physical and mental health. Technology-based dementia education has been noted to have equivalent effects as in-person education, but with the added benefit of asynchronous and/or remote delivery, which increases accessibility. Using Cochrane review methodology, this study systematically reviewed the literature on technology-based dementia education and its impacts on caregivers. Technology-based delivery included dementia education delivered via the Internet, telephone, telehealth, videophone, computer, or digital video device (DVD). In the review, twenty-eight studies were identified with fourteen included in a meta-analysis, and these data revealed a significant small effect of technologically based dementia education on reducing caregiver depression, and a medium effect on reducing caregiver distress in response to caregivers' observations of behavioral problems displayed by persons with dementia. No evidence was found for a significant effect of the educational intervention on caregiver burden or self-efficacy, which are known to be gendered aspects of caregiving. None of the studies included in the meta-analysis reported separate outcomes for male and female care providers, which has implications for gendered caregiving norms and aspects of care.

**Registration number:** PROSPERO 2018 CRD42018092599.

## Introduction

Timely delivery of dementia diagnosis, care strategies, supports, and management planning can greatly improve health outcomes for persons with dementia [1, 2]. Persons with dementia

published in PlosOne. We have provided enough information for someone to replicate the review and to analyze our findings from this review.

**Funding:** Funding was provided by the Canadian Consortium on Neurodegeneration in Aging (CCNA). CCNA is supported by a grant from the Canadian Institutes of Health Research with funding from several partners including the Saskatchewan Health Research Foundation, the Centre for Aging and Brain Health, and the Alzheimer Society of Canada (ASC). The funder Caregiver Crosswalk Inc provided support in the form of salaries for author CW, but did not have any additional role in the study design, data collection and analysis, decision to publish, or preparation of the manuscript. The specific roles of these authors are articulated in the 'author contributions' section.

**Competing interests:** One author has a commercial affiliation no other authors have a conflict to declare. The funder Caregiver Crosswalk Inc provided support in the form of salaries for author CW, but did not have any additional role in the study design, data collection and analysis, decision to publish, or preparation of the manuscript. The specific roles of these authors are articulated in the 'author contributions' section. This does not alter our adherence to PLOS ONE policies on sharing data and materials.

experience progressive worsening of cognition and independent functioning, necessitating assistance with activities of daily life [3]. Informal caregivers, typically family members or friends, are involved in providing day-to-day over the course of the progression in symptoms of dementia, making their experience more intimate than that of healthcare professionals [4]. Informal caregivers report challenges in daily care; lack of external/community supports [3], difficulties associated with insufficient dementia knowledge and support [3], and barriers to obtaining and sharing of information [4], resulting in isolation in caregiving responsibilities [3]. Informal caregivers also face auxiliary challenges such as frustration, anxiety, depression, and burnout (i.e. state of prolonged stress; [3, 4]).

Lack of knowledge in one or more aspects of dementia care not only affects the mental health of caregivers [3, 4], but also their caregiving abilities [5–9]. Dementia care costs in Canada are estimated to be that of five and a half times higher for individuals diagnosed with dementia than those diagnosed with any other chronic condition [10]. The cost of care in Canada for Alzheimer and other dementias exceeds the cost of all other neurodegenerative conditions [11]. The bulk of this cost falls on the shoulders of informal caregivers and is represented in lost wages, and hours spent providing care at home [12]. Tasks of caregiving typically increase both in complexity and diversity, until eventually the informal caregiver assumes all household and financial aspects of care [12, 13]. Given the importance of informal care providers in provision of care, and the benefits of dementia education for care providers' care of themselves and the person living with dementia, it is paramount to provide more education on dementia for informal caregivers. Dementia education that is most helpful is comprised of strategies for care of close individuals who are diagnosed with dementia, knowledge about dementia course and progression, as well as self-care strategies and prevention of care burnout [3, 4].

Dementia education has been also noted to enhance care management and social support seeking behaviour by informal caregivers (e.g., seeking counselling, joining discussion forums, and informal support groups; [4]. Disparities in levels of formal education among higher- and lower-educated caregivers can also impact the quality of care [14, 15], but provision of dementia education can reduce knowledge disparities among higher- and lower-educated caregiver groups [6]. Other barriers to caregiving include insufficient information about behavioural symptoms of dementia and accurate understanding of the benefits of timely diagnosis [16]. Studies have shown benefits to care partners' caregiving abilities when interventions include psychoeducation [17], and are targeted at increasing communication between the informal caregiver and caree [18]. Overall, dementia education for informal caregivers seems to have a positive influence on both the care and the health outcomes of persons with dementia [6, 8, 19, 20].

Benefits of dementia education are well supported [4, 8, 16–18], and informal caregivers indicate receptiveness to a variety of educational modes, including community-based and internet-based resources [16] and online portals [21]. Internet-based interventions have been shown to improve caregiver confidence [22–25], reduce perceived burden [26], increase self-efficacy [21], decrease anxiety [21], decrease feelings of depression [26, 27], and increase dementia knowledge [28]. A recent systematic review of psychoeducational interventions for caregivers of people with dementia who live at home included studies that delivered technology-based interventions and found that the technology-based interventions reduced caregiver burden. [29]. Some examples of technology-based interventions for caregivers include websites [27], tailored web-based interventions with expert recommendations [30], hybrid interventions that include an online component and an interactive telephone component (e.g., dementia care consultation; [26], and interventions with a telephone-only component [31]. Studies comparing dementia education versus participation in a network of caregivers found that

interventions containing both an educational component [32, 33] and an interactive component had the greatest benefit and impact on skills [25], stress [23], attitudes [21], knowledge [28], self-efficacy [24], and perceived caregiver burden [26]. Interactive components of interventions, featuring discussion boards [27], email [27, 30], and multiple forms of informational presentation (e.g., video, case-based vignettes) [22, 32, 34], were the most highly rated [22, 27, 30, 31, 34].

## Digital educational tools for caregivers

Digital modes of education, such as knowledge conveyed via electronic devices, media, internet, closed-circuit computer networks, or web-based platforms, have an equivalent impact on learning as face-to-face modes [35]. Additionally, such learning offers an advantage of remote or asynchronous delivery, where each participant progresses through educational material in a self-paced manner [36, 37]. Asynchronous digitally based learning holds additional benefits for individuals living in remote or rural areas, where long wait times for specialized services and a lack of information/local educational programs are common [38]. Recent changes within health services and efforts to move services to virtual methods due to the COVID-19 pandemic, have seen online-based learning gain a new importance [39, 40]. Searching and summarizing existing research on current modes of digital education for informal care providers can serve as an important step toward implementing digital dementia education programs for caregivers and enhancing access to educational and support resources.

Past systematic reviews have examined internet-based interventions for caregivers of persons with dementia [41], focusing on evaluation methods, including qualitative interviews, satisfaction-based interviews, organizational feedback [42], as well as interventions delivered via telephone [43], and computer [43]. The present review examined the effect of digital dementia education delivered via technology with remote delivery capabilities, including smart phone applications, telephone, video, computer (online and network), and focused on studies with pre and post measures in order to more accurately determine the impact of digital education. Increased availability and ownership of smart devices with access to the internet have changed digital delivery of information within the past decade, as well as motivation to engage in education [44, 45], in part due to increased focus on remote and asychronous approaches for education necesited by the pandemic. Consequently, the current review explores the evidence base for different technologies used for dementia education with a focus on the quality of evidence and the assessed to provide this evidence, with an emphasis on different domains of care partner mental health.

## Method

### Study objectives and search strategy

The review used reporting guidelines from Preferred Reporting Items for Systematic Reviews and Meta-Analyses (PRISMA) [46], and Cochrane guidelines for systematic reviews and meta-synthesis [47, 48]. Risk of bias for each domain of studies was evaluated by two reviewers independently using the Cochrane handbook, and an overall risk of bias was determined based on the least favourable assessment across the domains [49]. The review addressed the following questions: 1.) What learning technologies, devices, online resources, and digital education tools on dementia are available for caregivers? 2.) How have these educational technologies been evaluated? 3.) What are the outcomes for informal caregivers? Databases were limited to the University databases, which included PsychINFO, MEDLINE, Cumulative Index of Nursing and Allied Health Literature (CINAHL), Sociological Abstract, AgeLine, and Embase. These databases were selected in consultation with a research librarian (for the syntax used in

the search strategy, see S1 Appendix). The current review includes the initial literature search, which was performed in 2018, and an update which was performed in 2020. The initial review and the update were performed by the same two reviewers and following the same data screening, extraction, and analysis.

## Data screening, extraction and analysis

Two reviewers independently screened titles, abstracts, and full text of systematically obtained studies. Reasons for exclusion of studies were recorded in the PRISMA diagram shown in Fig 1, and included a focus on a different population, no educational intervention offered, or no outcome measures reported. When studies contained multiple intervention groups, only studies with the group of interest (caregivers of persons with dementia) were retained. In cases where studies reported caregivers as a subset of a larger population of interest, data on caregivers were reported separately. Disagreement on study inclusion was resolved verbally, with a third individual serving as an adjudicator during lack of consensus. Data extraction was performed independently by two reviewers (AS and AD). Adapted and customized data extraction forms that included study details such as author, publication year, country, reasons for inclusion or exclusion, design, methods, duration of intervention, participant characteristics, outcomes, and risk of bias (composed at individual and aggregate levels for studies included in the meta-analysis) [47, 48]. Studies were managed, stored, and analyzed with Review Manager [50]. Meta-synthesis was also performed with RevMan for the studies using identical outcomes and measures. This process was repeated in its entirety for the review update (Fig 2).

## Study inclusion and exclusion criteria

Studies were peer-reviewed, available via the University library electronic database, and included at least one technology or digital tool (defined below) used to convey dementia education which was targeted to informal caregivers (e.g., partners, family friends, or adult children of persons living with dementia). The search criteria and databases were selected in consultation with a research librarian, specializing in social sciences. The selection of databases was based on the relevance to the topic of investigation and availability of published studies via the university search engines. We decided to focus on studies published in English language due to limitations in understanding of other languages by the two reviewers. No date limits were applied to the search. Digital tools included e-learning and other electronic media, intranet, teleconferencing networks, and telephones. Digital delivery did not have to be online, for example internal resources available via computer networks within organizations were also considered. Lastly, selected studies contained interventions with pre- and post- outcome measures. Excluded studies were other systematic reviews, studies with only a face-to-face educational component, studies targeting participants other than caregivers of persons with dementia, studies where care recipients were diagnosed with conditions other than dementia, and studies where dementia education did not constitute a substantial portion of the intervention.

## Risk of bias and study quality

Risk of bias was evaluated with Cochrane Collaboration's Risk of Bias tool as low, medium, high, or unknown [48]. Quality of the findings from studies included in the meta-analysis were evaluated using Grading of Recommendations, Assessment, Development and Evaluations (GRADE) [51]. Quality of finding from studies not included in the meta-analysis was assessed by adapting the checklist from the Critical Appraisal Skills Programme [52], where the range of possible quality scores for studies ranged from 0 to 21 for studies using a

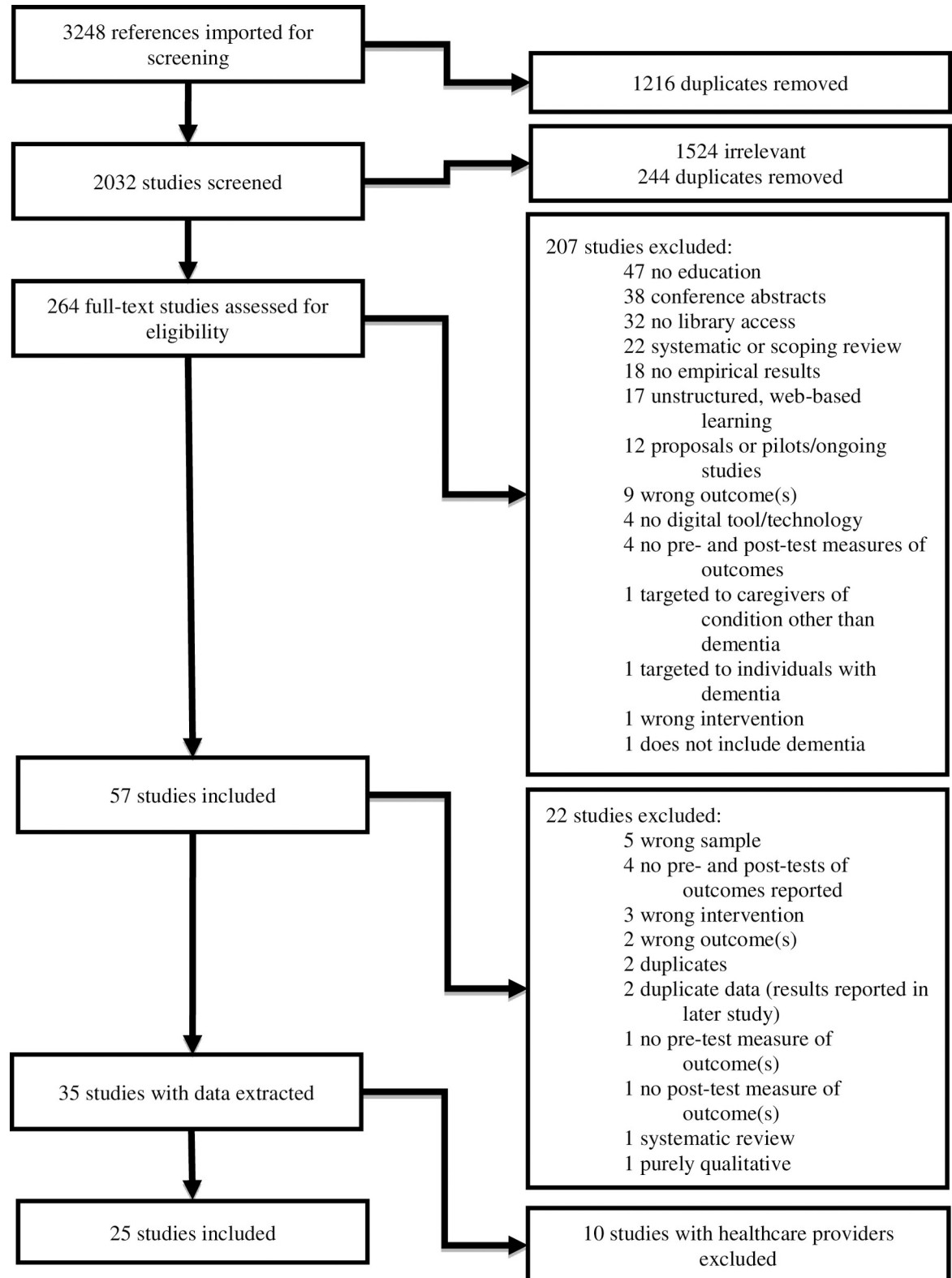

**Fig 1. Preferred reporting items for systematic reviews and meta-analysis (2018) (Moher et al.).**

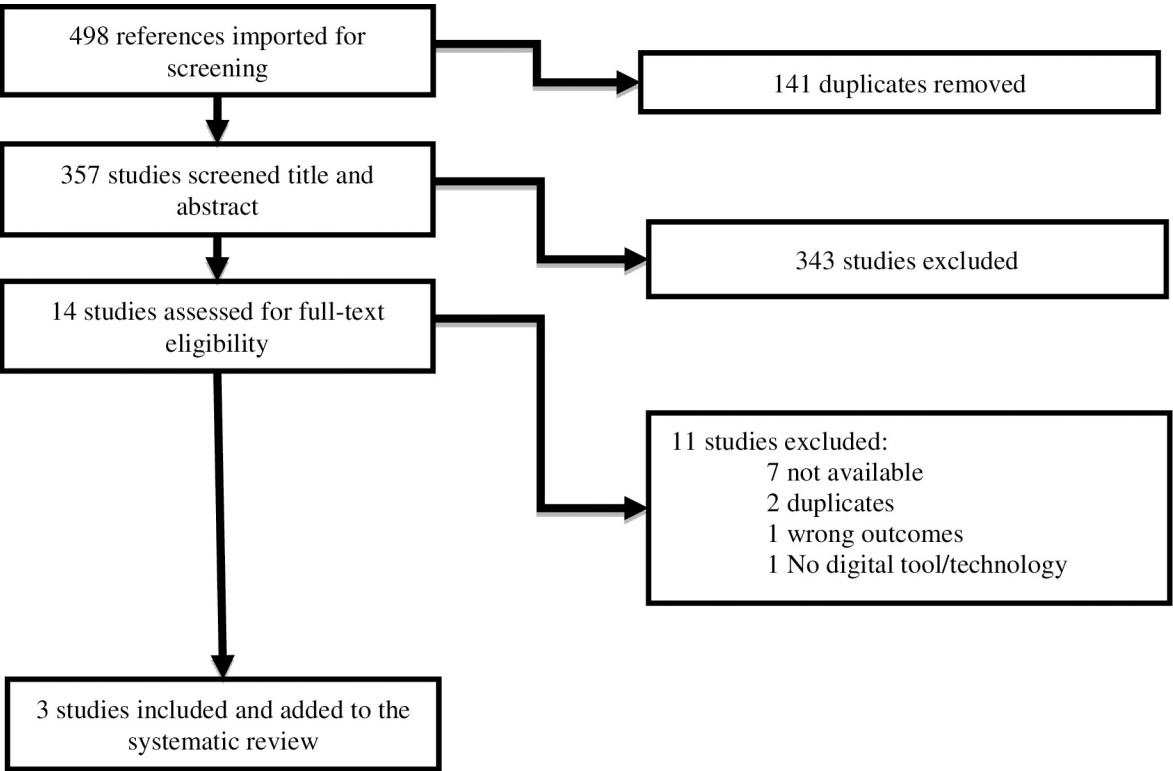

**Fig 2. Preferred reporting items for systematic reviews and meta-analysis update.**

randomized controlled trial design, and 0 to 18 for studies using a single group with a pre- and post-intervention measure (see S1 and S2 Tables). Risk of bias and quality assessment was performed independently by two reviewers (AS and AD). Disagreement on risk of bias or quality assessment rating was resolved verbally, with a third individual (MEO) serving as an adjudicator during lack of consensus.

## Results

### Study demographics

A total of 28 studies were included in the review. Of these 14 used the RCT design, equivalent scales and outcomes and were included in a meta-analysis (see Table 1). The remaining 14 studies included a mixture of RCT ($n = 6$) and pre- and post-intervention design ($n = 8$) and were described qualitatively (see Table 2). The 6 RCT studies excluded from the meta-analysis did not use equivalent scales and outcomes. Studies were published between 1990 and 4 October 2020 and included a technology-based dementia education component (see Tables 1 and 2). The number of participants in studies ranged from $N = 8$ to $N = 290$, and the percentage of female caregivers ranged from 63% to 91%. Study duration ranged between 17 days to 1 year, and intervention follow-up ranged from immediate post-intervention, up to 12 months post-intervention.

Of the total 28 studies, some of the studies were conducted in the United States ($n = 22$), Netherlands ($n = 3$), Poland/Spain/Denmark ($n = 1$), France ($n = 1$), and China ($n = 1$) (see S3 Table). In 82% of the studies, the authors did not report whether each study was conducted

**Table 1. Review studies included in meta-analysis.**

| Author, year | Theoretical model | Design, sample size, attrition, female (%) | Duration of type of intervention | Follow-up | Outcome scales |
|---|---|---|---|---|---|
| Beauchamp, 2005 | Stress Process Theory [62], Theory of Reasoned Action [57], Transactional Model of Behavior Change [58]. | RCT, $n$ = 150 (treatment group), $n$ = 149 (waitlist control), $n$ = 30, Female: 73% | 30 days, 3 online modules focusing on problem-focused techniques and social support skills. | Baseline, 30 days | CSS (subscales), CSI, PAC, CES-D |
| Boots, 2018 | The Stress Process Model [62] and Bandura's Self-Efficacy Model [59] | RCT, $n$ = 31(intervention), $n$ = 37 (waitlist control), $n$ = 13, Female: 78% | 8 weeks, face-to-face intake session, tailored online thematic modules, focusing on psychoeducation, behavioral modeling, reflective assignments, change plans, and email feedback from the coach, face-to-face evaluation session | Baseline, post-intervention | CSES, CES-D, PMS, HADS-A, ICECAP-O, GDS |
| Czaja, 2013 | No model | RCT, $n$ = 38 (intervention), $n$ = (attention control), $n$ = 36, n = 36 (information only control) $n$ = 17, Female: 81% | 5 months, 6, 1-hour monthly sessions, 4 educational seminars, delivered via videophone, 5 videophone support sessions | Baseline, 5 months | CES-D, RMBPC, Social support, PAC |
| Gallagher-Thompson, 2010 | No model | RCT, $n$ = 40 (skills training intervention), $n$ = 36 (education DVD control), $n$ = 24, Female: 87% | 12 weeks, 2.5 hours, CBT skills training program delivered via DVD | Baseline, 6 weeks | CES-D, RMBPC |
| Glueckauf, 2007 | No model | RCT, $n$ = 12 (intervention group), $n$ = 8 (routine education/support control), $n$ = 22, Female: 71% | 12 weeks, 12 weekly telephone-delivered educational sessions | Baseline, 1-week post-treatment | CAI, CSES-R, CES-D, ISS, IFS, ICS |
| Kajiyama, 2013 | No model | RCT, $n$ = 75 (treatment group), $n$ = 75 (education/information condition control), $n$ = 47, Female: 84% | 3 months, 6 online modules | Baseline, post-intervention | PSS, RMBPC, CES-D, PQoL |
| Martindale-Adams, 2013 | No model | RCT, $n$ = 75 (treatment group), $n$ = 75 (print materials control), $n$ = 15, Female: 83.75% | 1 year, 12, 1-hour sessions on education, and skills-building, delivered via telephone | Baseline, 6 months, 12 months | ZBI, CES-D, GWBS, RMBPC |
| Núñez-Naveira, 2016 | No model | RCT, $n$ = 31 (usual lifestyle control), $n$ = 30 (treatment group), $n$ = 16, Female: 63% | 3 months, 5 online modules delivered via Smartphone or tablet, covering topics of care and management | Baseline, post-intervention | ZBI, GDS, CES-D, CCS, RCSS |
| Cristancho-Lacroix, 2015 | Stress Process Theory [62], Bandura's Self-Efficacy Model [59] | RCT, $n$ = 24 (usual care control), $n$ = 17 (treatment group), $n$ = 7, Female: 78% | 3 months, weekly web-based psycho-educational sessions lasting 15 to 30 minutes | Baseline, 3 months, 6 months | PSS, CSES-R, RMBPC, ZBI, BDI-II), Self-Perceived Health |
| Finkel, 2007 | No model | RCT, $n$ = 12 (basic education control), $n$ = 13 (treatment group), $n$ = 21, Female: 69% | 6 months, 12 educational sessions, 8 hours in average duration, delivered via videophone | Baseline, 6 months | CES-D, RMBPC, CHHBS, SSS |
| Gant, 2007 | No model | RCT, $n$ = 17 (video/teleconference condition), $n$ = 15 (education condition), $n$ = 4, Female: 0% | 10 video sessions on caregiving strategies, weekly telephone calls from a coach | Baseline, post-intervention | CSES-R, PNAS, RMBPC, |
| Steffen, 2000 | No model | RCT, $n$ = 10 (video series), $n$ = 9 (classroom viewing), $n$ = 9 (waitlist/control condition), $n$ = 5, Female: 75.8% | 8 weeks, once a week, psychoeducational session with a 30-minute video segment. | Baseline, post-intervention | CgAI, BDI-II, CSES-R |
| Steffen, 2016 | No model | RCT, $n$ = 23 (behavioral coaching), $n$ = 35 (basic education), $n$ = 31, Female: 100% | 14 weeks, 10, 30-minute video segments (DVD), 10 weekly telephone calls, 2 maintenance calls | Baseline, post-intervention, 6 months | BDI-II, RMBPC, NAS, MAACL-R (subscales) |

*(Continued)*

**Table 1.** (Continued)

| Author, year | Theoretical model | Design, sample size, attrition, female (%) | Duration of type of intervention | Follow-up | Outcome scales |
|---|---|---|---|---|---|
| Kwok, 2013 | Psychosocial Transition and Stress Coping Theory [60, 61] | RCT, $n$ = 20 (remote video group), $n$ = 18 (treatment group), $n$ = 4, Female: 71% | 12 weeks, 12, 30-minute psychoeducational sessions delivered via telephone, DVD containing educational information | Baseline, post-intervention, 3 months | ZBI, CSES-R |

RCT = randomized controlled trial

AD = Alzheimer's Disease

CSS = Coping Skills Scale

CSI = Caregiver Strain Instrument

PAC = Positive Aspects of Caregiving

CES-D = Center for Epidemiologic Studies Depression Scale

RMBPC = Revised Memory and Behavior Problems Checklist

CAI = Caregiver Appraisal Inventory

CSES = Caregiving Self-Efficacy Scale

CSES-R = Caregiving Self-Efficacy Scale-Revised

ISS = Issue Severity Scale

IFS = Issue Frequency Scale

ICS = Issue Change Scale

PSS = Perceived Stress Scale

PMS = Pearlin Mastery Scale

PQoL = Perceived Quality of Life

GWBS = General Well-Being Scale

GDS = Global Deterioration Scale

HADS = Hospital and Anxiety Depression Scale

ICECAP-O = the Investigating Choice Experiments for the Preferences of Older People Capability measure for Older People

CCS = Caregiver Competence Scale

RCSS = Revised Caregiving Satisfaction Scale

BDI-II = Beck Depression Inventory II

CHHBS = Caregiver Health and Health Behaviors Scale

SSS = Social Support Scale

PNAS = Positive and Negative Affect Scale

CgAI = Caregiver Anger Interview

NAS = Negative Affect Scale

MAACL-R = Multiple Affect Adjective Check List Revised.

with rural or urban populations. A single study reported using a mixed sample (rural and urban), one study reported using a rural population, and three studies used urban populations. The majority of participants in the studies were reported either as a spouse, partner, a significant other ($n$ = 15), or an adult child ($n$ = 11).

Several studies ($n$ = 8) used a theoretical model to conceptualize their interventions. These models were Borkman's Model [53] ($n$ = 1), emphasizing peer support among caregivers, Medical Model of Dementia [54] ($n$ = 1), International Classification of Functioning, Disability and Health Model [55], Adaptation Coping Model [56] ($n$ = 1), Theory of Reasoned Action [57] ($n$ = 1), Transactional Model of Behavior Change [58] ($n$ = 1), Bandura's Self-Efficacy Model [59] ($n$ = 2), Psychosocial Transition and Stress Coping Theory [60]; [61] ($n$ = 1), and the Stress Process Model ($n$ = 5) [62]. One study used a model developed specifically for the delivered intervention, the *CARES* for Families model [63] ($n$ = 1).

**Table 2. Review studies not included in meta-analysis.**

| Author, year | Theoretical model | Design, sample size, attrition, female (%) | Duration of type of intervention | Follow-up | Outcome scales |
|---|---|---|---|---|---|
| Goodman, 1990 | Borkman's Model [53] | RCT, $n$ = 22 (network lecture sequence), $n$ = 18 (lecture network sequence), $n$ = 41, Female: 75% | 3 months, 12 phone-delivered lectures where participants could call at any time | Baseline, 3 months, 6 months | MPBC, ZBI, MHI, knowledge of AD |
| Czaja, 2018 | Stress Process Theory [62] | Pre and post measures, $n$ = 146, Female: 76% | 6 months, 12, 60-min individual (6 face-to-face and 6 telephone) skill building and educational sessions and 5 telephone support groups | Baseline, 6 months, 12 months | CES-D, BDI, SSS, CSES, SSQ, PACS, QLS |
| Wijma, 2018 | No model | Pre and post measures, $n$ = 42, Female: 77% | 3 weeks, virtual reality 360 degrees simulation video featuring three scenarios, 3, 20-minute e-lessons | Baseline, 3 weeks | ADQ, IRI, PIC, TOA, DRS |
| Goodman & Pynoos, 1990 | No model | RCT, $n$ = 35 (lecture component), $n$ = 31 (network component), $n$ = 15, Female: 77% | 12 weeks, 12 telephone accessed taped lectures about Alzheimer's disease over a 12-week period | Baseline, 12 weeks | ZBI, Mental health, CERS, Social support measure, Networking measure, Information about AD |
| Hattink, 2015 | Medical Model of Dementia (DSM-IV, 2004), International Classification of Functioning, Disability and Health Model [55], Adaptation Coping Model [56] | RCT, $n$ = 27 (group began program right away), $n$ = 32 (group began program after 4 months), $n$ = 59, Female: 49% | 4 months, 8 online modules relating to different topics on dementia and dementia care | Baseline, 4 months | ADKS, Attitudes regarding dementia- ADS |
| Hicken, 2017 | No model | RCT, $n$ = 77 (internet group), $n$ = 78 (telephone group), $n$ = NA, Female: 90% | 4–6 months, weekly delivery of educational information using the Internet or a home telehealth device | Baseline, post intervention | ZBI (short form), MARWIT (short form), PHQ, DIS |
| Mavandadi, Wray, et al., 2017 | No model | RCT, $n$ = 140 (enhanced caregiver services), $n$ = 290 (clinical assessment and referral), Female: 73% | 3 months, module calls delivered 2 to 3 times a month, each 45 to 60 minutes in duration. | baseline, 3 months, 6 months | ZBI (short form), NPI, RMBPC |
| Mavandadi, Wright, et al. 2017 | No model | RCT, $n$ = 31 (dementia care management, $n$ = 25 (usual care), $n$ = 32, Female: 97.3% | 3 months, 7 online modules | baseline, 3 months, 6 months | ZBI, NPI, RMBPC, Management of Meaning, LCAS |
| Davis, 1998 | No model | Pre and post measures, $n$ = 17, $n$ = 5, Female: 75% | 12 weeks, 2-hour in home training session, 12, 45–60 minute structured phone calls | Baseline, 12 weeks | RMBPC, RPSI, USS, CLS |
| Gaugler, 2015 | *CARES* for Families [63]. | Pre and post measures, $n$ = 41, $n$ = NA, Female: 90% | 17.05 days, online, 1 hour informational modules. | Baseline, post intervention | Dementia Care Knowledge |
| Easom, 2013 | No model | Pre and post measures, $n$ = 85, $n$ = 76, Female: 78% | 6 months, telephone support group sessions on topics concerning communication and caregiving | Baseline, 6 months | RAA, ZBI, CES-D, RMBPC, CSES-R, DIS |
| Glueckauf, 2004 | No model | Pre and post measure, $n$ = 20, $n$ = 9, Female: 90% | 4 months, series of six 45-minute live online, interactive classes | Baseline, 4 months | CSES, SRGS, CAI |
| Griffiths, 2018 | Stress Process Theory [62] | Pre and post measure, $n$ = 57, $n$ = 7, Female: 91% | 6 weeks, 75 minute teleconferences led by instructors, 5–15 videos with educational messages | Baseline, 6 weeks to 1 months after completion | ZBI, CES-D, PCMS, RMBPC |

*(Continued)*

**Table 2.** (Continued)

| Kajiyama, 2018 | No model | Pre and post measure, $n = 19$, $n = 6$, Female: 76% | 4 weeks, 18 Webnovella episodes, 15–20 minutes in duration | Baseline, 4 weeks | PSS, CES-D, Knowledge Survey |

RCT = randomized controlled trial

NA = not available, AD = Alzheimer's Disease

MPBC = Memory and Problem Behavior Checklist

BDI = Overall Burden Interview

ZBI = Zarit Burden Interview

MHI = Mental Health Index

CERS = Caregiver Elder Relationship Scale

ADQ = Approach to Dementia Questionnaire

IRI = Interpersonal Reactivity Index

PIC = Pressure from Informal Care

TOA = Trust in Own Abilities

DRS = Dyadic Relationship Scale

ADKS = Alzheimer's Disease Knowledge Scale

ADS = Approaches to Dementia Scale

MARWIT = Marwit-Meuser Caregiver Grief Inventory

SSS = Social Support Scale

QLS = Quality of Life Scale

SSQ = Social Support Questionnaire

PHQ = Patient Health Questionnaire

DIS = Desire to Institutionalize Scale

NPI = Neuropsychiatric Inventory Questionnaire

PACS = Positive Aspects of Caregiving Scale

RMBPC = Revised Memory and Behavior Problems Checklist

LCAS = Lawton Caregiving Appraisal Scale

RPSI = Rational Problem Solving Inventory

USS = Use of Social Support

CLS = Caregiver Life Satisfaction

RAA = Risk Appraisal Assessment

CES-D = Center for Epidemiologic Studies Depression Scale

CSES-R = Caregiving Self-Efficacy Scale-Revised

CSES = Caregiving Self-Efficacy Scale

SRGS = Stress-Related Growth Scale

CAI = Caregiver Appraisal Inventory

PCMS = Pearlin Caregiver Mastery Scale

PSS = Perceived Stress Scale

## Risk of bias

Risk of bias (ROB) was assessed at the study level (see method) and aggregately for studies included in the meta-analysis. The majority of studies had an unknown risk of bias due to a lack in reporting of randomized allocation ($n = 15$). Many did not report concealment of allocated condition ($n = 17$), blinding of participants to condition ($n = 15$) and condition at outcome assessment levels ($n = 21$), and thus received an unknown ROB rating. Due to the nature of treatment (participation in an educational intervention), it is likely that blinding was not possible in all cases. Some studies reported incomplete outcome data or only significant outcomes ($n = 7$) and were thus rated as having a high ROB. Many reported complete outcome data and appeared free of selective reporting ($n = 17$), and thus received a low ROB rating (see S4 and S5 Tables).

## Study quality

GRADE was used to assess quality of findings from studies included in the meta-analysis at the outcome level for measures of depression (measured by the Center for Epidemiological Studies Depression Scale [CES-D] or the Beck Depression Inventory [BDI-II]), behavioral problems, caregiving self-efficacy, caregiver burden, and positive caregiving experience (S6 Table). Of the studies included in the meta-analysis, twelve studies included some measure of depression. Depression (measured by the BDI-II), caregiving self-efficacy, caregiver burden, and positive caregiving experience was rated at a low-level enhancing confidence that the true effect lies closely to estimate of the effect. These studies had smaller effect sizes with larger confidence intervals. Outcomes such as behavioral problems measured by the Revised Memory and Behavior Problem Checklist (RMBPC) and (BDI-II) were rated as very low, expressing limited confidence in the effect estimate, due to concerns about heterogeneity of results, and a high risk of publication bias.

The quality of the studies excluded from the meta-analysis was performed with CASP criteria, allowing for quality evaluation at each study level (see S1 and S2 Tables). RCT studies ($n = 6$) were rated as high quality (score range between 13 and 18), and non-RCT studies ($n = 8$) were all rated as medium quality (score range from 8 to 14), with the exception of two studies, which were rated as high (score range from 15 to 21). CASP quality ratings of medium to high indicated that study authors measured the intended constructs in their intervention assessments.

## Characteristics of learning technologies

Telephone-based dementia education was the most common intervention modality ($n = 18$), followed by video-based learning ($n = 18$), and online-based learning ($n = 11$) (see S7 Table). One study used a novel approach with virtual reality to stimulate the experience of an individual diagnosed with dementia. Studies also used a variety of methods to deliver educational content including learning modules ($n = 8$), a website platform ($n = 1$), and a live online class featuring an instructor ($n = 2$). A few telephone-based interventions used a telephone support group ($n = 3$). One live version of a telephone-delivered lecture included a live instructor who delivered education ($n = 1$), and two studies featured pre-taped lectures which were accessible via telephone. Mixed modes of telephone and visual content were delivered via devices such as a videophone ($n = 1$), computer-telephone integration system ($n = 1$), and Smartphone or tablet device ($n = 1$). Video-based learning included supplementary visual resources featuring role playing and modeling via storytelling ($n = 2$). Other supplementary resources were textual resources such as printed materials and workbooks, immediate feedback, exercises, knowledge tests, and references to other online resources (reported in 12% of studies).

## Evaluation of digital educational tools

All of the studies ($n = 28$) used scales to assess the outcomes of dementia knowledge, caregiver burden/distress, competence in aspects of care, supports, management, health of the caregiver, and satisfaction with caregiving (see Tables 1 and 2). Some of the more commonly used scales by the studies were the Revised Memory and Behavior Problems Checklist (RMBPC) for measure of observable behavioral problems in dementia patients [64]. The Beck Depression Inventory (BDI-II) [65] and the Center for Epidemiologic Studies Depression Scale (CES-D) [66], measured depression. Caregiver burden was measured by scales such as the Zarit Burden Inventory (ZBI) [67], and the Revised Caregiving Self Efficacy Scale (CSES-R) [68] was used to measure cognitive and behavioral approaches in managing care of persons with dementia. For the remaining scales, please refer to Tables 1 and 2.

## Focus of educational content

The number of sessions varied from 3 to 24. The intervention was typically distributed between 17 days and 12 months, with an individual duration of 15 and 75 minutes, and involved various modes of delivery such as online, videophone, telephone, smartphone, virtual reality, and DVDs. The focus of the content revolved around three main areas; enhancing dementia knowledge strategies, management of daily tasks, and self-management: support-seeking and self-care.

**Enhancing dementia knowledge.** Twelve studies included a dementia knowledge component [23, 32, 69–79]. Areas of interest under strategies were problem-solving strategies, specifically related to aspects of care [75, 77, 80–82], and communication strategies with the care recipient [23, 25, 26, 71, 74, 77, 81, 83, 84].

**Management of daily tasks.** Management of daily tasks included themes such as creating caregiving goals [26, 32, 74–76, 78, 82, 85, 86], managing medical aspects of care [31, 32], managing financial and legal aspects of care [31, 32, 74, 77, 83], managing home safety [32, 77], and behavior management [32, 71, 76, 78, 81, 84, 86–89].

**Self-management.** The umbrella of self-management included categories of education targeted toward support seeking, which included sub-themes of social support seeking [70, 71, 74–76, 79, 85], family support seeking [26], and seeking of respite through community resources [32, 70, 81, 83, 90]. Under this same umbrella, a sub-category targeted toward self-care included themes of enhancing coping strategies [25, 80, 88], personal health management [76–78, 83], personal mood management [90], enhancing self-care [23, 71, 75, 90], and increasing self-efficacy in care [84, 85]. In summary, education was targeted toward many aspects of personal care. Information regarding care strategies was delivered with an even balance of strategies for care provision and external support attainment. It can be argued that educational content was comprehensive and addressed all aspects of personal and care provider well-being.

## Meta-analysis

Meta-analysis was performed using Review Manager Software, where the main outcomes were calculated as an overall effect size for each of the outcomes. Some of the outcomes were measured by different versions of the same scales (full versus shortened versions), thus standardized mean differences (SMD) with 95% confidence intervals (CI) were used to estimate pooled effect sizes for each outcome. Depression was measured separately and pooled by measures such as Center for Epidemiologic Studies Depression Scale (CES-D) and Beck Depression Inventory, Second Edition (BDI-II). Caregiver distress in response to observed behavioral problems was measured by the Revised Memory and Problem Behavior Checklist (RMBPC). Caregiving self-efficacy, or the ability to respond to disruptive behaviors and control upsetting thoughts was measured by the Caregiving Self-Efficacy Scale (CSES). Positive caregiving experience was measured by the Caregiver Appraisal Inventory (CAI). Caregiver perception of burden was measured by the Zarit Burden Interview (ZBI). Results were obtained after pooling each outcome in the meta-analyses.

**Depression.** In seven studies measuring depression with the CES-D, there were statistically significant differences in depression between treatment and control groups (SMD -0.11; 95% CI -0.25 to 0.02) (Fig 3), indicating no effect of dementia education on depression. However, in three additional studies measuring depression with the BDI-II, there was a statistically significant difference between the intervention and control groups (SMD -0.99; CI -1.39 to -0.58) (Fig 4), indicating another small effect of dementia education on depression. We performed a sensitivity analysis by removing lower quality studies, and found the difference was no longer statistically significant.

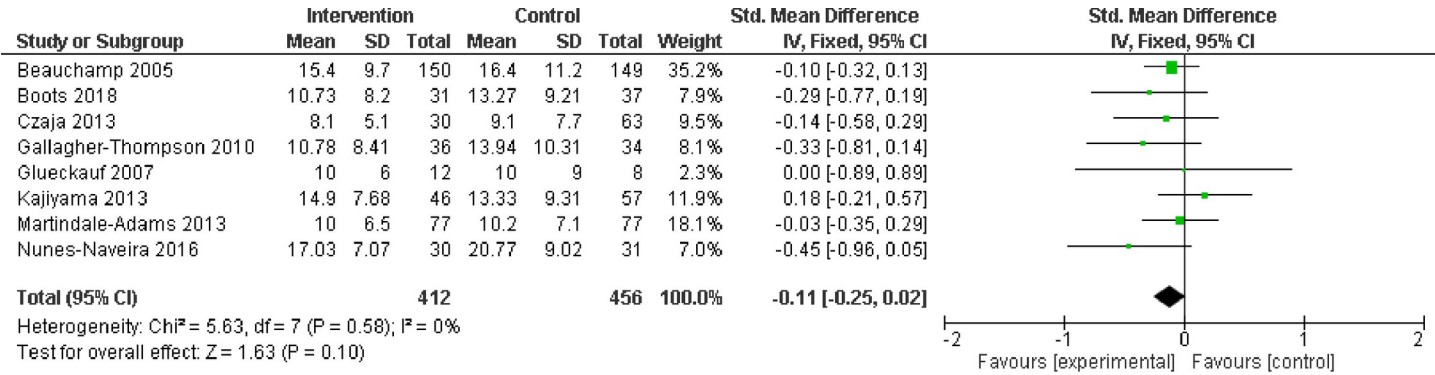

**Fig 3. Depression assessed with: Center for Epidemiological Studies Depression Scale (CES-D).**

**Observable behavioral symptoms.** In eight studies measuring caregiver response to observed behavioral problems with the RMBPC, there was a significant difference (SMD -0.44; CI -0.61 to -2.27) (Fig 5), indicating a medium effect of dementia education on observable behavioral symptoms, a finding that was robust to a sensitivity analysis removing lower quality studies.

**Caregiving self-efficacy.** In the six studies measuring caregiving self-efficacy with the CSES, the control groups appeared slightly more favoured than the intervention (SMD 0.21; CI -0.03 to 0.44) (Fig 6), indicating dementia education does not positively impact caregivers' ability to respond to behavioral symptoms or control upsetting thoughts (similar findings were found when removing lower quality studies from the meta-analysis).

**Caregiver perceived burden.** Six studies measuring caregiver burden with ZBI, found no significant differences between the intervention and control groups (SMD 0.05; CI -0.20 to 0.30) (Fig 7), indicating no effect of dementia education on perceived burden (studies included in the meta-analysis were of similar quality with none removed in the sensitivity analysis).

**Positive caregiving experience.** Three studies measuring positive caregiving experience indicated no significant difference between intervention and control groups (SMD -0.04; CI -0.23 to 0.15) (Fig 8).

## Gender and sex in reporting of study outcomes

A particularly interesting finding in this review is that none of the studies (both excluded and included in the meta-analysis) reported separate outcomes for male and female caregivers. A substantial body of literature indicates that male and female caregivers, particularly spousal care providers, report different levels of caregiver strain [91–96], different needs for attainment of emotional supports [97–103], and quality of life in relation to their caregiving role [104–106]. Female care providers tend to report more symptoms of depression, as well as higher

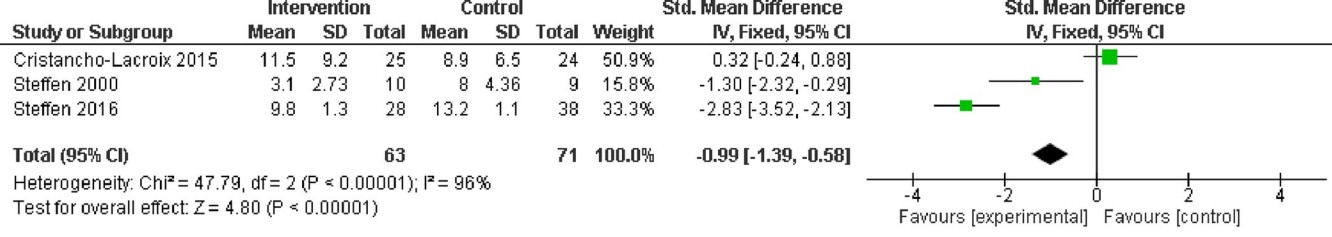

**Fig 4. Depression assessed with: Beck Depression Inventory (BDI-II).**

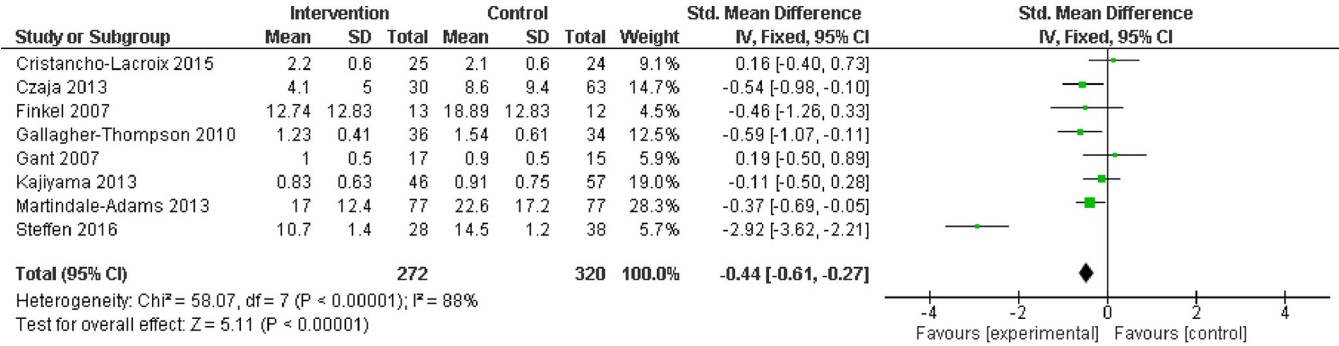

**Fig 5. Behavioral problems assessed with: Revised Memory and Behavior Problem Checklist (RMBPC).**

physical and emotional burden than male care providers [94]. Female care providers also report feeling more obligated to fulfill a wider variety of roles including physical care and emotional support [106]. Additionally, female care providers are typically viewed as natural and preferred caregivers due to cultural norms [95, 107].

While women report higher emotional strain than men [108], women are also more likely to seek out social supports [98]. And while women report greater emotional strain and symptoms of depression [93], men are more likely to process emotional and psychological effects of their role individually without sharing of feelings with others [97, 98]. Therefore, men and women differ in the way they approach their caregiver role as well as in the ways they cope [91]. This is particularly meaningful because a 10 year risk of Alzheimer's Disease was found to be greater for APOE ε44 genotype carriers [109] which is more predictive of AD when associated with cerebrovascular disease, and for women, middle to late adulthood hypertension was associated with an increased risk of dementia [110, 111]. Taking into consideration the female gendered roles in care provision, and the likelihood that female spouses are at a higher risk of dementia, it would be important and relevant to target interventions and report separate outcomes for male and female care providers.

## Discussion

Studies included in this review used a variety of methodological approaches, scales, and measured outcomes; and due to this variation, a meta-analysis was only possible for five outcomes. The findings from the meta-analysis indicate a small effect of interventions on reduced depression, which is consistent with other literature indicating a similar significant but small effect

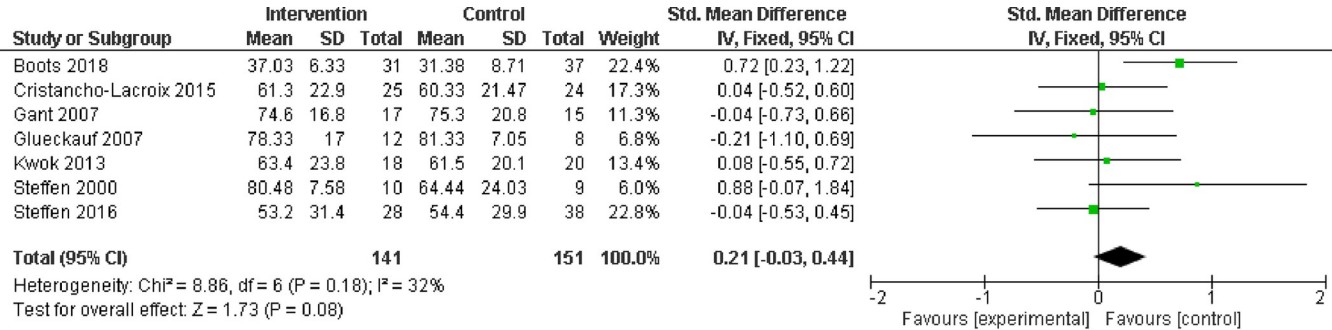

**Fig 6. Caregiving self-efficacy assessed with: Caregiving Self-Efficacy Scale (CSES).**

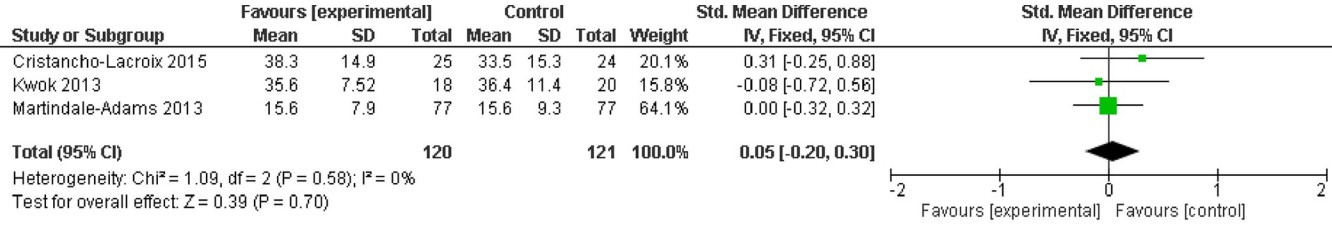

**Fig 7. Caregiver burden assessed with: Zarit Burden Interview (ZBI).**

[4, 112–115]. There was also a medium effect of interventions on caregiver distress in response to behavioral symptoms. These results are consistent with other literature indicating a significant effect of dementia education on reduced caregiver stress [112]. It is notable, however, that no significant effect was demonstrated for reducing caregiver perceived burden or for increasing self-efficacy, or for increasing the positive aspects of caregiving. While caregiver burden and self-efficacy are broad and difficult constructs to measure [114, 116], it is possible that they were not consistently captured across studies, thus an absence of a significant effect. Moreover, many of the studies did not adequately report randomization, blinding of participants, or blinding at the outcome assessment level. Removal of lower quality studies resulted in similar findings, but the significant but small impact on depressive symptoms became statistically non-significant. The results from this review are further limited by the focus on the English language literature, the inability to tease out which digital technologies might be more impactful than others, and are limited in their applicability to rural/remote dwelling caregivers due to lack of data, and are unclear in how well these results translate to males and females due to lack of sex, gender, or sex and gender stratification.

Previous literature indicates mixed findings for the effects of dementia education on self-reported caregiver burden, with some authors noting positive effects on burden [4, 117–119], and some indicating no effect [112, 114, 116], thus supporting mixed findings to support for the effectiveness of educational, digitally based interventions on self-reported caregiver burden and self-efficacy. Factors which may impact the effect of education on caregiver burden are the type of intervention, where gain-focused re-appraisal strategies may have a greater effect on burden than education alone for example, allowing to create concrete goals to alleviate specific burdens [113]. Interventions that are higher in intensity and duration may also exert a greater effect on burden [117]. Other possible contributing factors may be the timing of follow-up, where significant effects diminish over time [118]; and point in progression of dementia, where the effect of an intervention on burden may be higher for caregivers of persons with dementia earlier in the progression of dementia, than later [120]. Caregiver burden findings are consistent with the noted variations in effects in previous literature, and likely due to similar alterations in structure and design of previous studies.

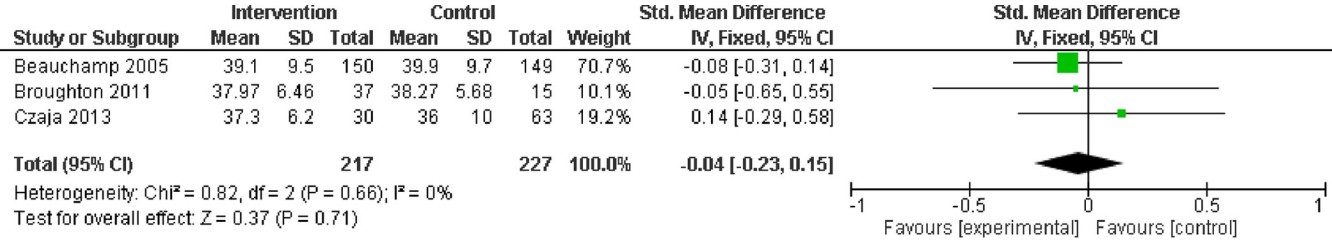

**Fig 8. Positive caregiving experience assessed with: Positive Aspects of Caregiving (PAC).**

Meta-analysis also showed no effect of the educational intervention on caregiving self-efficacy. These results could be consistent with variation in caregiver reports in self-efficacy, where caregivers note a positive change after treatment, but not necessarily a statistically significant change from the control group [114]. The positive change may also vary by duration of follow-up, where longer time to follow-up leads to a reduction in effects [118]. Caregiver self-efficacy and burden are large, multidimensional constructs for a variety of aspects of dementia care. Self-efficacy encompasses the ability to obtain respite, respond to a range of behavioral symptoms and control upsetting thoughts, while burden encompasses depression, mental health, negative affect, social support, stress, and coping.

A 2020 review [29] that included technology based interventions and in-person group interventions might reveal factors underlying these contradictory data across studies on the impact of education for dementia. In this review of English and Spanish language literature [29] technology-based and in-person group based psychoeducation interventions had different impacts for caregivers of persons living at home with dementia. Notably, the technology interventions reduced burden whereas the in-person interventions had broder impacts including on reducing mental health concerns, reducing burden, and increasing self-efficacy. It is possible that the convience of technology based educational interventions only served to reduce burden wheras the relationships formed in the group setting and sense of universality in the caregiving experience helped more broadly [121]. The recent review [29] that appears to contrast technology-based and in-person group educational interventions was non-equivalent–technology was telephone based or asyncrhonous and completed in solitidue. Group based educational sessions could also serve as support groups [121], which are beneficial for caregivers whether they are in-person or digitally delivered. Comparisons of digitially mediated and in-person intervnetions via RCT are needed.

One of the limitations of the current review is that not all digitally based learning is likely created equally; for example, the studies reviewed in [29] found broad positive impacts for in-person group interventions, which we postulate could be due to the potential for unintended support group-like side-effects of a group psychoeducation intervention. Arguably a digitally based psychoeducation group that mimicked in-person interaction, such as one by group videoconferencing could confer the same benefits. For the studies included in the current review digitally based learning occurred via a variety of methods, including online-learning, telephone-based learning, and video-based learning. Studies using mixed modes of educational delivery (e.g., video, case-based learning, role playing, discussion platforms, and periodic knowledge tests) were also used to deliver education to caregivers. The qualitative analysis revealed digitally based interventions, focused on enhancing dementia knowledge, care abilities, behavior management, caregiver distress, skills/goals, and coping strategies. Some interventions also focused on obtaining specific supports such as community resources, medical care, legal and financial advice, home safety, and social support seeking. Finally, interventions also focused on enhancing aspects of caregiver health such as healthy lifestyle, mood management, self-care, relationships, and self-efficacy. Future work matching benefits for caregivers with specific methods for digitally based education for dementia care is needed; for example, are group-based activities more effective at producing broad spectrum positive outcomes for caregivers? Are these more effective if the digital delivery more closely matches in-person group interactions; for example, we would postulate that videoconferencing would create more meaningful group cohesions than would telephone group educational interventions. This foundational work is needed to better understand how digital education should be designed to be optimal.

An additional limitation of this review is the focus on all-cause dementia or any cause of dementia. Different dementia presentation might impact caregivers differently and, therefore,

have different implications for dementia education. Moreover, different outcomes of dementia education focussed on generic aspects of caregiving might not be useful for persons caring for persons with atypical dementia presentations. For example, the needs of care partners of persons with dementia due to frontotemporal lobar degeneration appear unique [121].

This review is notable for what it found, but also for gaps that became apparent when examining the body of literature on digitally delivered education for dementia. Despite the clear advantages of using digitally based education on dementia for those who reside in rural locations, the majority (82%) did not specify the living demographics of their caregiver populations (e.g., rural versus urban living). Urban and rural populations significantly differ in their ability to access and secure external supports and services [38, 122], thus such a distinction would have been useful in determining the effect of the intervention when access to services is limited. Caregivers from rural areas face unique challenges in care due to a lack of access of community-based resources, respite programs, and specialist care (physiotherapy, occupational therapy, counseling etc.) [38, 123]. The ability to secure such supports could influence chosen outcomes, therefore it would be beneficial to determine service accessibility and barriers, before employing certain measures. Future studies could address specific populations (e.g., rural), and design their digitally based dementia education interventions for such populations. Studies aimed at digitally based dementia education for rural caregivers would be important because the busy daily care workload could effectively reduce time to participate in any kind of dementia education [124].

A second notable gap that became apparent after reviewing this body of literature was the lack of attention to sex and gender. None of the studies reported separate outcomes for males and females, nor adapted aspects of their interventions for male and female care provider needs respectively. This gap in the literature is perplexing given the evidence suggesting male and female care givers differ in their care giving roles, with a greater emphasis on gendered care roles [91, 98, 107], and in ways in which they cope, for example, women are more likely to seek external resources and respite [97, 104]. Previous research has shown that female caregivers report higher levels of emotional strain and burden [100, 125], and that male caregivers seek less social supports [91, 92, 94]. Male and female caregivers, particularly spouses of individuals with dementia, report differing levels of caregiver strain, where female caregivers report higher levels of strain [91, 92, 94], and female caregivers are more likely to seek out emotional support than male caregivers [98]. These findings are important because females have a higher chance of being an APOE 4 genotype carrier which when combined with cerebrovascular disease (also higher among women in later age) [109], is predictive of higher likelihood of Alzheimer's Disease (AD) in women [110, 111] making spousal males more likely to become caregiver. Also, these findings are relevant because studies report significantly higher depression in female than male caregiver counterparts and higher care-related burden [125–127]. Male and female caregivers also differ in their approach toward caregiving tasks; they approach caregiving from different coping strategies (e.g., emotionally-focused feminine orientations versus masculine task-oriented approaches) which has implications for emotional strain reported by female caregivers [128]. Together these findings play a role in future of caregivers of individuals with dementia, were males will be likely to be engaged more in care giving roles, which will in turn necessitate greater effort to support these individuals with greater shift in access toward resources and supports which would alleviate support seeking burden [96]. Digitally-delivered education for these individuals should be targeted toward finding ways to increase self-efficacy toward seeking external and social supports in caregiving [91, 92, 94]. Also, finding ways to support male and female caregivers, may prove to be necessary, as the pattern in needs is notably different; for example, female caregivers need more support toward preventing caregiver strain and burnout [100, 125] and depression [125–128], but generally

receive less informal support, as female caregivers are typically perceived as "natural caregivers", thus are thus perceived as needing less help [100, 129]. Thus targeting interventions for female caregivers could include more ideas for seeking social support and transcending gender-biased views suggesting that they need less support [125, 129]. Similarly, for male caregivers, introducing content that provides ideas on how to seek emotional support from family, friends, and available services, may serve as a protective factor in continuing to seek help and supports and counteract the found tendency of male caregivers to engage in help seeking [129]. While the included studies did not separate their outcomes by gender, it would have been useful to know how male and female care providers responded, and what were some of the potential gaps in the approach of each type of educational intervention?

## Conclusion

The systematic review findings and the meta-analysis indicate that digitally based dementia education can have a positive effect on caregiver depression, ability to manage behavioral issues in caregiving, and can improve the caregiving experience. The focus of sessions in the included digitally delivered interventions (in the present review) cover a wide area of topics including dementia knowledge, ability to provide care, behavior management, coping strategies, access to supports such as community resources, medical care, legal and financial advice, enhancing home safety, support seeking, and other aspects of caregiver health such as mood management, self-care, and sense of self-efficacy. One of the implications of this review is the efficacy of digitally-based education that is asynchronous. Asynchronous delivery would be possible to integrate dementia education within busy daily caregiving duties. Another implications of this review is the efficacy of the remote education for care partners. The present time of pandemic and isolation necessitates more efforts to deliver education remotely, and while remote education has been demonstrated to have similar effects as face-to-face education [22–25], the urge to deliver interventions remotely has never been greater. An emphasis on remote delivery for interventions is helpful for rural families whose access to supports, services, and interventions is markedly limited [38, 122]. Accessibility of interventions for rural dwelling families of persons with dementia is one core aspect of health equity for rural dementia care. The authors of the present study outline and stress the urge in accelerating efforts to deliver remote digitally based dementia education to caregivers.

## Supporting information

**S1 Checklist. PRISMA 2009 checklist.**
(DOC)

**S1 Appendix. Search syntax.**
(PDF)

**S1 Table. Adapted quality rating criteria for randomized controlled trial studies (studies not included in meta-analysis).**
(PDF)

**S2 Table. Adapted quality rating criteria for non-randomized controlled trial studies (studies not included in meta-analysis).**
(PDF)

**S3 Table. Characteristics of included studies.**
(PDF)

**S4 Table. Risk of bias assessment for RCT studies not included in meta-analysis.**
(PDF)

**S5 Table. Risk of bias assessment for RCT studies included in meta-analysis.**
(PDF)

**S6 Table. Quality of studies included in meta-analysis conveyed with Grading of Recommendations, Assessment, Development and Evaluations (Pourahmadi et al.).**
(PDF)

**S7 Table. Teaching and learning approaches.**
(PDF)

## Author Contributions

**Conceptualization:** Andrea Scerbe, Megan E. O'Connell, Arlene Astell.

**Data curation:** Andrea Scerbe, Andrea DesRoches.

**Formal analysis:** Andrea Scerbe, Andrea DesRoches.

**Methodology:** Andrea Scerbe, Andrea DesRoches.

**Project administration:** Andrea Scerbe.

**Resources:** Megan E. O'Connell.

**Writing – original draft:** Andrea Scerbe.

**Writing – review & editing:** Andrea Scerbe, Megan E. O'Connell, Arlene Astell, Debra Morgan, Julie Kosteniuk, Ivan Panyavin, Andrea DesRoches, Claire Webster.

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
