## [Decision Letter · Decision Letter 0]

22 Jun 2021

PONE-D-21-03124

Digital Tools for Delivery of Dementia Education for Caregivers of Persons with Dementia: A Systematic Review and Meta-Analysis

PLOS ONE

Dear Dr. O'Connell,

Thank you for submitting your manuscript to PLOS ONE. After careful consideration, we feel that it has merit but does not fully meet PLOS ONE’s publication criteria as it currently stands. Therefore, we invite you to submit a revised version of the manuscript that addresses the points raised during the review process.

In addition, please address the following:

1) PRISMA is a systematic review reporting recommendation, not a recommendation for systematic review conduct

2) there is not a standard Cochrane data extraction form. Please clarify the source of this form or omit the mention

3) Cochrane risk of bias is conducted for each domain of study conduct. How did the authors evaluate overall risk of bias for the studies?

4) GRADE is not used for the quality of studies but rather the quality (or confidence) of effect estimates. Please correct the use of GRADE in this paper.

5) Funnel plots for assessing the possibility of publication bias are not appropriate with fewer than at least 10 studies.

We look forward to receiving your revised manuscript.

Kind regards,

Lisa Susan Wieland

Academic Editor

PLOS ONE

Journal Requirements:

"The authors have declared that no competing interests exist. "

We note that one or more of the authors are employed by a commercial company: Caregiver Crosswalk Inc.

2.1. Please provide an amended Funding Statement declaring this commercial affiliation, as well as a statement regarding the Role of Funders in your study. If the funding organization did not play a role in the study design, data collection and analysis, decision to publish, or preparation of the manuscript and only provided financial support in the form of authors' salaries and/or research materials, please review your statements relating to the author contributions, and ensure you have specifically and accurately indicated the role(s) that these authors had in your study. You can update author roles in the Author Contributions section of the online submission form.

2.2. Please also provide an updated Competing Interests Statement declaring this commercial affiliation along with any other relevant declarations relating to employment, consultancy, patents, products in development, or marketed products, etc.  

Reviewers' comments:

Reviewer's Responses to Questions

**Comments to the Author**

1. Is the manuscript technically sound, and do the data support the conclusions?

Reviewer #1: Yes

Reviewer #2: No

2. Has the statistical analysis been performed appropriately and rigorously? 

Reviewer #1: Yes

Reviewer #2: Yes

3. Have the authors made all data underlying the findings in their manuscript fully available?

Reviewer #1: Yes

Reviewer #2: Yes

4. Is the manuscript presented in an intelligible fashion and written in standard English?

Reviewer #1: Yes

Reviewer #2: Yes

5. Review Comments to the Author

Reviewer #1: A well-structured and thorough review, of clinical and social importance. Your focus on digital technologies seems particularly apt in current COVID-19 circumstances, and very applicable to the current environment of healthcare delivery. Some structural changes and better situation of your discussion within the current research would improve the applicability of this review outside of your own working context.

• I would like to see justification of why only English-language studies were included, and see this addressed under limitations.

• A critical and structured reflection on your review of the strengths and limitations of your study should be included at the beginning of the discussion.

• In general, I think your discussion and clinical implications could have a wider scope. Though it’s great that you spoke with a consultant about this work, I don’t think it should be the main basis of this section of your discussion.

• I would like to see your discussion better situated within current research and clinical practice outside of this one clinical setting to improve the scope and impact of this review.

• Your findings around gendered caregiving are really interesting, and I think you can make more of these in the discussion and recommendations for future research and policy.

• You need a proper conclusion that brings together all elements of the paper, outside of the views of authors and one consultant.

Reviewer #2: Studying digital delivery of interventions for carers is very timely given the global pandemic. As the authors state, there is benefit to carers being able to access remote and asynchronous interventions. The authors also raise an excellent point that research points to male-female differences in coping with caring, such that we might expect different responses to interventions depending on coping style. Unfortunately, however, the paper tries to cover too much ground and does not quite cover any of it satisfactorily given the mismatch between aims and reported findings. There are very many reviews of non-pharmacological interventions for family carers of people with dementia and a review now needs to be excellent quality if it is to be worth publishing. The following major revisions would be required if this review is to usefully add to the literature:

1) Awareness of relevant recent reviews

The authors should update their background literature to include citations relevant recent reviews, especially those that include a technology component e.g.

Frias, C. E., Garcia‐Pascual, M., Montoro, M., Ribas, N., Risco, E., & Zabalegui, A. (2020). Effectiveness of a psychoeducational intervention for caregivers of people with dementia with regard to burden, anxiety and depression: a systematic review. Journal of advanced nursing, 76(3), 787-802.

2) Clarity of aim(s) &associated presentation of mateerial

The stated aims of this study would be appropriate for a scoping review (what tech available, how is it evaluated, which outcomes?). However, for a meta-analysis, there should be a clear question that would best be answered through use of meta-analysis. If the authors wish to focus on whether results are reported separately according to gender or sex this should be part of the aim; conversely, if there is to be a ‘focus on studies with pre-post measures’ (p. 7) or ‘focus on motivation and accessibility’ (p. 7), then findings relating to these issues should be provided. The authors should re-write the paper so that the aims and content are consistent with each other.

3) Appropriateness of methods

3.1 Please ensure that there is a match between the study selection (based on eligibility criteria) and the proposed synthesis strategy (in this case, meta-analysis). Single group pre-post studies are not relevant to a meta-analysis of RCTs

3.2 Please include search dates & explain why there were two searches with two separate PRISMA flow-charts.

4) Results

4.1 At present, there is a good deal of space taken up reporting excluded studies. N=28 studies were deemed eligible of which 14 were included in the meta-analysis.

4.2 Reporting of results should be checked for accuracy. Specifically:

- the self-efficacy analysis indicates that control is favoured over intervention with an effect size of .21 (CI -0.03, 0.44) rater than the reported ‘no effect’. Please check and revise paper accordingly.

- the PAC analysis indicates no effect, but this is reported as ‘positive effect on PAC’ (p. 17, p. 19). Please check and revise paper accordingly.

- Table 8 – number of participants differs between forest plots and table 8 for CES-D (depression) and self-efficacy

4.3 Missing – summary of ‘control’ conditions

5) Discussion

- Please compare findings to those of other relevant reviews & consider why there may be discrepancies

- Please add a ‘limitations’ section with appropriate critical appraisal of the methodology used e.g. inclusion of ‘live’ phone interventions when stated interest in ‘asynchronous’ interventions

- p. 21 to 23 appears to be a report of a discussion with a consultant on the impact of COVID-19 on carers. I’m assuming that this paper has been submitted elsewhere previously & it was suggested that ‘clinical implications’ were considered? It would be helpful if the findings from this useful chat were distilled to the points most pertinent to the review.

6. PLOS authors have the option to publish the peer review history of their article (what does this mean?). If published, this will include your full peer review and any attached files.

Reviewer #1: **Yes: **Emily West

Reviewer #2: No

---

## [Author Response · Author response to Decision Letter 0]

3 Nov 2021

Lisa Susan Wieland

Academic Editor

PLOS ONE

Thank you for providing us with this opportunity to revise the manuscript. WE WERE UNABLE TO UPDATE THE ORDER OF FILES FOR THE RESUBMISSION!

In addition, please address the following:

1) PRISMA is a systematic review reporting recommendation, not a recommendation for systematic review conduct

Thank you for pointing out our sloppy use of language – we have addressed this error in the revised manuscript. 

2) there is not a standard Cochrane data extraction form. Please clarify the source of this form or omit the mention

Thank you we have deleted the reference to a standard form. 

3) Cochrane risk of bias is conducted for each domain of study conduct. How did the authors evaluate overall risk of bias for the studies?

We have added the citation to the Cochrane training handbook. Higgins JPT, Savović J, Page MJ, Elbers RG, Sterne JAC. Chapter 8: Assessing risk of bias in a randomized trial. In: Higgins JPT, Thomas J, Chandler J, Cumpston M, Li T, Page MJ, Welch VA (editors). Cochrane Handbook for Systematic Reviews of Interventions version 6.2 (updated February 2021). Cochrane, 2021. Available from www.training.cochrane.org/handbook. We also specified that when we reported on the risk of bias for the result from each study this was based on the least favourable assessment across the domains considered as per Higgins et al., 2021 on page 7 of the revised manuscript. 

4) GRADE is not used for the quality of studies but rather the quality (or confidence) of effect estimates. Please correct the use of GRADE in this paper.

Thank you for pointing out our sloppy use of language – we have addressed this error in the revised manuscript. 

5) Funnel plots for assessing the possibility of publication bias are not appropriate with fewer than at least 10 studies.

Thank you for pointing out our error and we have deleted the two funnel plots.

Reviewers' comments:

Reviewer's Responses to Questions

Comments to the Author

1. Is the manuscript technically sound, and do the data support the conclusions?

Reviewer #1: Yes

Reviewer #2: No

2. Has the statistical analysis been performed appropriately and rigorously? 

Reviewer #1: Yes

Reviewer #2: Yes

3. Have the authors made all data underlying the findings in their manuscript fully available?

Reviewer #1: Yes

Reviewer #2: Yes

4. Is the manuscript presented in an intelligible fashion and written in standard English?

Reviewer #1: Yes

Reviewer #2: Yes

5. Review Comments to the Author

Reviewer #1: A well-structured and thorough review, of clinical and social importance. Your focus on digital technologies seems particularly apt in current COVID-19 circumstances, and very applicable to the current environment of healthcare delivery. Some structural changes and better situation of your discussion within the current research would improve the applicability of this review outside of your own working context.

• I would like to see justification of why only English-language studies were included, and see this addressed under limitations.

We have included the reasoning regarding our justification in the paper e.g., limitations – inability to speak or understand other languages by the reviewers, also limitations of available university search engine, thus publications in English only (pg. 8-9).

• A critical and structured reflection on your review of the strengths and limitations of your study should be included at the beginning of the discussion.

We agree and we have made this change in the revised manuscript – please see the first paragraph of the revised discussion on p. 17 and 18. 

• In general, I think your discussion and clinical implications could have a wider scope. Though it’s great that you spoke with a consultant about this work, I don’t think it should be the main basis of this section of your discussion.

We have revised the discussion and hopefully have sufficiently broadened the scope. We focused on limitations in the review and gaps identified to expand on future directions that could have clinical implications on pages 20-23.

• I would like to see your discussion better situated within current research and clinical practice outside of this one clinical setting to improve the scope and impact of this review.

We agree, and have linked the findings to the current research and to recent reviews in the similar area (see discussion pages 18-20. We discussed the clinical implications in the prior point, but refer you to pages 20-23 of the revised discussion. 

• Your findings around gendered caregiving are really interesting, and I think you can make more of these in the discussion and recommendations for future research and policy.

Thank you we agree, and we have tried to incorporate gendered caregiving and sex issues more into our discussion on pages 21-23.

• You need a proper conclusion that brings together all elements of the paper, outside of the views of authors and one consultant.

We agree, and we have added a conclusion section in the revised manuscript on pages 23-24. 

Reviewer #2: Studying digital delivery of interventions for carers is very timely given the global pandemic. As the authors state, there is benefit to carers being able to access remote and asynchronous interventions. The authors also raise an excellent point that research points to male-female differences in coping with caring, such that we might expect different responses to interventions depending on coping style. Unfortunately, however, the paper tries to cover too much ground and does not quite cover any of it satisfactorily given the mismatch between aims and reported findings. There are very many reviews of non-pharmacological interventions for family carers of people with dementia and a review now needs to be excellent quality if it is to be worth publishing. The following major revisions would be required if this review is to usefully add to the literature:

1) Awareness of relevant recent reviews

The authors should update their background literature to include citations relevant recent reviews, especially those that include a technology component e.g.

Frias, C. E., Garcia‐Pascual, M., Montoro, M., Ribas, N., Risco, E., & Zabalegui, A. (2020). Effectiveness of a psychoeducational intervention for caregivers of people with dementia with regard to burden, anxiety and depression: a systematic review. Journal of advanced nursing, 76(3), 787-802.

We have made more of an effort to include existing and relevant reviews with a technology-based component, including the recommended one above. Thank you for noting this. This is briefly discussed in the introduction, but is a focus in the discussion on pages 19-20.

2) Clarity of aim(s) &associated presentation of mateerial

The stated aims of this study would be appropriate for a scoping review (what tech available, how is it evaluated, which outcomes?). However, for a meta-analysis, there should be a clear question that would best be answered through use of meta-analysis. If the authors wish to focus on whether results are reported separately according to gender or sex this should be part of the aim; conversely, if there is to be a ‘focus on studies with pre-post measures’ (p. 7) or ‘focus on motivation and accessibility’ (p. 7), then findings relating to these issues should be provided. The authors should re-write the paper so that the aims and content are consistent with each other.

Although the questions asked were broad and seem more characteristic of a scoping review, we wanted use these to extract the relevant information pertaining to our singular question; what is the existing evidence for digitally-based dementia interventions for caregivers with pre-post measures. In effort to clarify this, we changed the writing to reflect our question. As noted, systematic reviews usually are written very precisely to reflect specific key elements of a review question; for intervention questions, these are the population, intervention, comparison, and outcome. We have attempted to clarify this more in the review. The intended aim of the present study was to carry out a systematic review and describe the data qualitatively. The meta-analysis was not planned originally, but with the consistency in used measures and outcomes in some of the studies, we decided to carry out the quantitative synthesis or the meta-analysis on an ad hoc basis. We clarified this more in the study, and appropriately changed the wording and the intent. Also, the separate reporting according to sex and gender was not a part of our original question, this is something that came out of the review, and thus we decided to discuss it ad hoc, as it seemed pertinent to modern caregivers, as well as the differences in gender, health and care. We have clarified this in the study as well on page 9.

3) Appropriateness of methods

3.1 Please ensure that there is a match between the study selection (based on eligibility criteria) and the proposed synthesis strategy (in this case, meta-analysis). Single group pre-post studies are not relevant to a meta-analysis of RCTs

A meta-analysis was not the intent nor focus of this review, it was performed ad hoc. The intent of this study was a systematic review of existing modes of remote to be described qualitatively thus we focused on digital education on dementia for caregivers with a single group pre-post studies. We have made an effort to clarify this more in the paper (pg. 9). 

3.2 Please include search dates & explain why there were two searches with two separate PRISMA flow-charts.

The two PRISMA flow charts correspond to the initial search which was performed in the previous year, the second PRISMA chart corresponds to the updated review which would capture any new studies within the year since the initial search. This reasoning is now included in the paper (pg. 9, 40, 41).

4) Results

4.1 At present, there is a good deal of space taken up reporting excluded studies. N=28 studies were deemed eligible of which 14 were included in the meta-analysis.

The excluded studies were also the focus of the review (contain existing modes of remote, digital education on dementia for caregivers with a single group pre-post studies thus the findings are deemed pertinent), and their data is accordingly qualitatively described. 

4.2 Reporting of results should be checked for accuracy. Specifically:

- the self-efficacy analysis indicates that control is favoured over intervention with an effect size of .21 (CI -0.03, 0.44) rater than the reported ‘no effect’. Please check and revise paper accordingly.

We have made the corrections as noted. 

- the PAC analysis indicates no effect, but this is reported as ‘positive effect on PAC’ (p. 17, p. 19). Please check and revise paper accordingly.

Thank you, we have made the suggested correction. 

- Table 8 – number of participants differs between forest plots and table 8 for CES-D (depression) and self-efficacy

Thank you for noting this, we have made the correction and the forest plots are deleted.

4.3 Missing – summary of ‘control’ conditions

We have added these summaries for each control condition that were missing from each study in Table 3.

5) Discussion

- Please compare findings to those of other relevant reviews & consider why there may be discrepancies

Thank you for your recommendation, we have included a section comparing our findings to other relevant reviews and why there might be discrepancies on pages 18-21.

- Please add a ‘limitations’ section with appropriate critical appraisal of the methodology used e.g. inclusion of ‘live’ phone interventions when stated interest in ‘asynchronous’ interventions

We did not wish to convey a sole interest in asynchronous interventions, merely digitally based interventions and have revised the abstract and study aims to make this more clear. 

- p. 21 to 23 appears to be a report of a discussion with a consultant on the impact of COVID-19 on carers. I’m assuming that this paper has been submitted elsewhere previously & it was suggested that ‘clinical implications’ were considered? It would be helpful if the findings from this useful chat were distilled to the points most pertinent to the review.

This was not an original consideration in the review, but was added to help tie the findings to existing clinical implications, and better situate our findings in context of real care. As recommended, we have decided to exclude this section as it did not seem pertinent to the review. We discuss the limitations of the review on page 18 and again on pages 20-23.

6. PLOS authors have the option to publish the peer review history of their article (what does this mean?). If published, this will include your full peer review and any attached files.

---

## [Decision Letter · Decision Letter 1]

25 May 2022

PONE-D-21-03124R1

Digital Tools for Delivery of Dementia Education for Caregivers of Persons with Dementia: A Systematic Review and Meta-Analysis

PLOS ONE

Dear Dr. O'Connell,

Thank you for submitting your manuscript to PLOS ONE. After careful consideration, we feel that it has merit but does not fully meet PLOS ONE’s publication criteria as it currently stands. Therefore, we invite you to submit a revised version of the manuscript that addresses the points raised during the review process.

Please respond to the two comments made by the reviewer. Identify the study as noted by the reviewer and check for other instances where the reference might be unclear. It is optional to address the second comment from the author, however others may also question it so consider addressing. 

We look forward to receiving your revised manuscript.

Kind regards,

Lisa Susan Wieland

Academic Editor

PLOS ONE

Journal Requirements:

Reviewers' comments:

Reviewer's Responses to Questions

**Comments to the Author**

1. If the authors have adequately addressed your comments raised in a previous round of review and you feel that this manuscript is now acceptable for publication, you may indicate that here to bypass the “Comments to the Author” section, enter your conflict of interest statement in the “Confidential to Editor” section, and submit your "Accept" recommendation.

Reviewer #3: All comments have been addressed

2. Is the manuscript technically sound, and do the data support the conclusions?

Reviewer #3: Yes

3. Has the statistical analysis been performed appropriately and rigorously? 

Reviewer #3: I Don't Know

4. Have the authors made all data underlying the findings in their manuscript fully available?

Reviewer #3: Yes

5. Is the manuscript presented in an intelligible fashion and written in standard English?

Reviewer #3: Yes

6. Review Comments to the Author

Reviewer #3: thank you for this much improved version i have two very minor comments:

Writing style, please ensure this is checked for example on page 35 the authors write "the studies reviewed in (29)" you need to refer to the name of the author here and not only the reference number or write in a previous review (29) ....

i don't feel too strongly but i would question the need for the following sentence to be included: "These findings are important because females have a higher chance of

being an APOE 4 genotype carrier which when combined with cerebrovascular disease (also

higher among women in later age) (109), is predictive of higher likelihood of Alzheimer’s

Disease (AD) in women (110, 111) making spousal males more likely to become caregiver."

7. PLOS authors have the option to publish the peer review history of their article (what does this mean?). If published, this will include your full peer review and any attached files.

Reviewer #3: No

---

## [Author Response · Author response to Decision Letter 1]

9 Jul 2022

Reviewers' comments:

Reviewer's Responses to Questions

Comments to the Author

1. If the authors have adequately addressed your comments raised in a previous round of review and you feel that this manuscript is now acceptable for publication, you may indicate that here to bypass the “Comments to the Author” section, enter your conflict of interest statement in the “Confidential to Editor” section, and submit your "Accept" recommendation.

Reviewer #3: All comments have been addressed

 We thank the reviewer for noting this. 

2. Is the manuscript technically sound, and do the data support the conclusions?

Reviewer #3: Yes

 We thank the reviewer for noting this.

3. Has the statistical analysis been performed appropriately and rigorously?

Reviewer #3: I Don't Know

4. Have the authors made all data underlying the findings in their manuscript fully available?

Reviewer #3: Yes

 We thank the reviewer for noting this.

5. Is the manuscript presented in an intelligible fashion and written in standard English?

Reviewer #3: Yes

 We thank the reviewer for noting this.

6. Review Comments to the Author

Reviewer #3: thank you for this much improved version i have two very minor comments:

Writing style, please ensure this is checked for example on page 35 the authors write "the studies reviewed in (29)" you need to refer to the name of the author here and not only the reference number or write in a previous review (29) ....

i don't feel too strongly but i would question the need for the following sentence to be included: "These findings are important because females have a higher chance of

being an APOE 4 genotype carrier which when combined with cerebrovascular disease (also

higher among women in later age) (109), is predictive of higher likelihood of Alzheimer’s

Disease (AD) in women (110, 111) making spousal males more likely to become caregiver."

We thank the reviewer for their comments, and we agree we needed an additional round of revisions to address the language, we added Frias and colleagues to the citation for the study previously identified only as 29. In addition, we have deleted the sentence suggested in the discussion.

These conform.

---

## [Decision Letter · Decision Letter 2]

5 Dec 2022

PONE-D-21-03124R2Digital Tools for Delivery of Dementia Education for Caregivers of Persons with Dementia: A Systematic Review and Meta-AnalysisPLOS ONE

Dear Dr. O'Connell,

Thank you for submitting your manuscript to PLOS ONE. After careful consideration, we feel that it has merit but does not fully meet PLOS ONE’s publication criteria as it currently stands. Therefore, we invite you to submit a revised version of the manuscript that addresses the points raised during the review process.

We look forward to receiving your revised manuscript.

Kind regards,

Ammal Mokhtar Metwally, Ph.D (MD)

Academic Editor

PLOS ONE

Additional Editor Comments:

Please note that your manuscript was reviewed by 2 experts in the field. There is consensus agreement that the idea of the article is interesting. Meanwhile, there is identified important problems in your submission and reviewers provided copious comments. Please consider responding to the reviewers’ remarks. The manuscript could be greatly strengthened by considering editing according to the specific mentioned comments.

Reviewers' comments:

Reviewer's Responses to Questions

**Comments to the Author**

1. If the authors have adequately addressed your comments raised in a previous round of review and you feel that this manuscript is now acceptable for publication, you may indicate that here to bypass the “Comments to the Author” section, enter your conflict of interest statement in the “Confidential to Editor” section, and submit your "Accept" recommendation.

Reviewer #3: All comments have been addressed

Reviewer #4: (No Response)

2. Is the manuscript technically sound, and do the data support the conclusions?

Reviewer #3: Yes

Reviewer #4: Partly

3. Has the statistical analysis been performed appropriately and rigorously? 

Reviewer #3: Yes

Reviewer #4: Yes

4. Have the authors made all data underlying the findings in their manuscript fully available?

Reviewer #3: Yes

Reviewer #4: Yes

5. Is the manuscript presented in an intelligible fashion and written in standard English?

Reviewer #3: Yes

Reviewer #4: Yes

6. Review Comments to the Author

Reviewer #3: thank you for this article i am happy that all queries have been responded to and address. a good article which is interesting to read.

Reviewer #4: 1. This manuscript has its clear good purposes for caregivers, i.e., analyzing impacts of technology-based dementia education on several life aspects (e.g., depression, distress, self-efficacy) of caregivers through meta-analysis. Regarding the present title, it could be a little vague for potential readers to quickly grasp what the main “outcomes” of interest (e.g., depression) that this paper would like to focus.

2. Introduction: Based on existing literature, it is evidently that digital modes of dementia have already been proven effective in terms of some aspects. And it is understandable that this study wanted to summarize relevant studies to serve as a step toward future education programs. However, walking through the Introduction section, it is still unclear: (1) in this area, what were the exact points which the past research has not been solved/answered; (2) so for this “pooled” study, what “expected” findings/knowledge different from those “single” studies in the past would bring into this area; and (3) what special view(s) that this review can give but past studies couldn’t? I think these should be said in a more systematic and deeper way in Introduction.

3. Study objectives and search strategy: I would like to suggest the three main questions this review addressed needing to have a strong link back to the Introduction. In other words, it would be better that each of them should have a background to let potential readers clearly understand why this question would be asked?

4. Study inclusion and exclusion criteria: dementia has different sub-types, so caregivers of persons with different sub-types may have different burdens. Because of this, even the same intervention may lead to different effectiveness, thus could be a reason biasing the “pooled” result. It is unclear that how the authors address this issue in searching, including, and excluding studies.

5. Meta-analysis: I would encourage to do some sensitivity analysis to test the stability and robustness of the “pooled” result, for example, study quality, etc.

6. Discussion: it would be better if this manuscript can strengthen the discussion regarding future dementia and education policy implications about this topic.

7. PLOS authors have the option to publish the peer review history of their article (what does this mean?). If published, this will include your full peer review and any attached files.

Reviewer #3: No

Reviewer #4: No

---

## [Author Response · Author response to Decision Letter 2]

19 Feb 2023

PONE-D-21-03124R2

Digital Tools for Delivery of Dementia Education for Caregivers of Persons with Dementia: A Systematic Review and Meta-Analysis

PLOS ONE

Dear Dr. Ammal Mokhtar Metwally, Ph.D (MD)

Academic Editor

PLOS ONE

We thank you for the opportunity to revise this manuscript further. We have addressed all reviewer comments below, and we believe the reviewer comments have contributed to an improved iteration of this manuscript. 

• An unmarked version of your revised paper without tracked changes. You should upload this as a separate file labeled 'Manuscript'

Comments to the Author

1. If the authors have adequately addressed your comments raised in a previous round of review and you feel that this manuscript is now acceptable for publication, you may indicate that here to bypass the “Comments to the Author” section, enter your conflict of interest statement in the “Confidential to Editor” section, and submit your "Accept" recommendation.

Reviewer #3: All comments have been addressed

Reviewer #4: (No Response)

2. Is the manuscript technically sound, and do the data support the conclusions?

Reviewer #3: Yes

Reviewer #4: Partly

Author’s response: We believe the sensitivity analysis suggested by a reviewer will help to reassure all regarding how the data support the conclusions. 

3. Has the statistical analysis been performed appropriately and rigorously?

Reviewer #3: Yes

Reviewer #4: Yes

4. Have the authors made all data underlying the findings in their manuscript fully available?

Reviewer #3: Yes

Reviewer #4: Yes

5. Is the manuscript presented in an intelligible fashion and written in standard English?

Reviewer #3: Yes

Reviewer #4: Yes

6. Review Comments to the Author

Reviewer #3: thank you for this article i am happy that all queries have been responded to and address. a good article which is interesting to read.

Author’s response: We thank reviewer 3 for noting that the revisions addressed the previous reviewers comments. 

Reviewer #4: 1. This manuscript has its clear good purposes for caregivers, i.e., analyzing impacts of technology-based dementia education on several life aspects (e.g., depression, distress, self-efficacy) of caregivers through meta-analysis. Regarding the present title, it could be a little vague for potential readers to quickly grasp what the main “outcomes” of interest (e.g., depression) that this paper would like to focus.

Author’s response: This was a helpful suggestion; we had revised the title a few rounds of revision ago to focus on specific outcomes. Digital tools for delivery of dementia education for caregivers of persons with dementia: A systematic review and meta-analysis of impact on caregiver distress and depressive symptoms

2. Introduction: Based on existing literature, it is evidently that digital modes of dementia have already been proven effective in terms of some aspects. And it is understandable that this study wanted to summarize relevant studies to serve as a step toward future education programs. However, walking through the Introduction section, it is still unclear: (1) in this area, what were the exact points which the past research has not been solved/answered; (2) so for this “pooled” study, what “expected” findings/knowledge different from those “single” studies in the past would bring into this area; and (3) what special view(s) that this review can give but past studies couldn’t? I think these should be said in a more systematic and deeper way in Introduction.

Author’s response: We agree with this reviewer – we needed to be clearer. The revised manuscript points out the links more clearly between the literature review, the main aims, and the objectives. 

3. Study objectives and search strategy: I would like to suggest the three main questions this review addressed needing to have a strong link back to the Introduction. In other words, it would be better that each of them should have a background to let potential readers clearly understand why this question would be asked?

Author’s response: We hope the revised introduction addresses this concern. 

4. Study inclusion and exclusion criteria: dementia has different sub-types, so caregivers of persons with different sub-types may have different burdens. Because of this, even the same intervention may lead to different effectiveness, thus could be a reason biasing the “pooled” result. It is unclear that how the authors address this issue in searching, including, and excluding studies.

Author’s response: This is an excellent point and we have added the fact that the literature review focussed on dementia due to any etiology as a limitation for our findings. 

5. Meta-analysis: I would encourage to do some sensitivity analysis to test the stability and robustness of the “pooled” result, for example, study quality, etc.

Author’s response: This is an excellent point, and we performed the below sensitivity analysis. We briefly described the sensitivity analysis in the revised results. 

Risk of bias improved after removal of Gallagher-Thompson, Glueckauf, Finkel, Kwok and Boots – for sensitivity analysis. 

 

New Figure; for sensitivity analysis 

Figure 1.3 Depression assessed with: Center for Epidemiological Studies Depression Scale (CES-D)

Note: In the above figure, Mean is the study mean for each group, SD is the standard deviation for each group, Total is the total number (N) in each group, Weight is the influence of studies on overall meta-analysis, Std. Mean Difference is the overall effect, Heterogeneity (I2) = 0%, p value indicates level of statistical significance.

 

Figure 1.5 Behavioral problems assessed with: Revised Memory and Behavior Problem Checklist (RMBPC)

Note: In the above figure, Mean is the study mean for each group, SD is the standard deviation for each group, Total is the total number (N) in each group, Weight is the influence of studies on overall meta-analysis, Std. Mean Difference is the overall effect, Heterogeneity (I2) = 88%, p value indicates level of statistical significance.

 

Figure 1.6 Caregiving Self-Efficacy assessed with: Caregiving Self-Efficacy Scale (CSES)

Note: In the above figure, Mean is the study mean for each group, SD is the standard deviation for each group, Total is the total number (N) in each group, Weight is the influence of studies on overall meta-analysis, Std. Mean Difference is the overall effect, Heterogeneity (I2) = 32%, p value indicates level of statistical significance. 

6. Discussion: it would be better if this manuscript can strengthen the discussion regarding future dementia and education policy implications about this topic.

Author’s response: We have added implications of these data in the revised discussion. 

7. PLOS authors have the option to publish the peer review history of their article (what does this mean?). If published, this will include your full peer review and any attached files.

Do you want your identity to be public for this peer review? For information about this choice, including consent withdrawal, please see our Privacy Policy.

Reviewer #3: No

Reviewer #4: No

---

## [Decision Letter · Decision Letter 3]

13 Mar 2023

Digital Tools for Delivery of Dementia Education for Caregivers of Persons with Dementia: A Systematic Review and Meta-Analysis

PONE-D-21-03124R3

Dear Dr. O'Connell,

We’re pleased to inform you that your manuscript has been judged scientifically suitable for publication and will be formally accepted for publication once it meets all outstanding technical requirements.

Kind regards,

Ammal Mokhtar Metwally, Ph.D (MD)

Academic Editor

PLOS ONE

Reviewers' comments:

Reviewer's Responses to Questions

**Comments to the Author**

1. If the authors have adequately addressed your comments raised in a previous round of review and you feel that this manuscript is now acceptable for publication, you may indicate that here to bypass the “Comments to the Author” section, enter your conflict of interest statement in the “Confidential to Editor” section, and submit your "Accept" recommendation.

Reviewer #3: All comments have been addressed

Reviewer #4: All comments have been addressed

2. Is the manuscript technically sound, and do the data support the conclusions?

Reviewer #3: Yes

Reviewer #4: Yes

3. Has the statistical analysis been performed appropriately and rigorously? 

Reviewer #3: N/A

Reviewer #4: Yes

4. Have the authors made all data underlying the findings in their manuscript fully available?

Reviewer #3: Yes

Reviewer #4: Yes

5. Is the manuscript presented in an intelligible fashion and written in standard English?

Reviewer #3: Yes

Reviewer #4: Yes

6. Review Comments to the Author

Reviewer #3: I would like to commend the authors for their patience in this review process - it seems you have had another reviewer added despite myself having approved your changes and the other original reviewer not responding. This should not have needed another reviewer. This is a good article

Reviewer #4: The comments have been addressed well by the authors and I have no further points to add into this manuscript.

7. PLOS authors have the option to publish the peer review history of their article (what does this mean?). If published, this will include your full peer review and any attached files.

Reviewer #3: No

Reviewer #4: No

---

## [Editor Report · Acceptance letter]

24 Mar 2023

PONE-D-21-03124R3 

Digital tools for delivery of dementia education for caregivers of persons with dementia: A systematic review and meta-analysis of impact on caregiver distress and depressive symptoms 

Dear Dr. O'Connell:

I'm pleased to inform you that your manuscript has been deemed suitable for publication in PLOS ONE. Congratulations! Your manuscript is now with our production department. 

Kind regards, 

on behalf of

Professor Ammal Mokhtar Metwally 

Academic Editor

PLOS ONE